# Role of BRAF in Hepatocellular Carcinoma: A Rationale for Future Targeted Cancer Therapies

**DOI:** 10.3390/medicina55120754

**Published:** 2019-11-21

**Authors:** Antonio Gnoni, Antonella Licchetta, Riccardo Memeo, Antonella Argentiero, Antonio G. Solimando, Vito Longo, Sabina Delcuratolo, Oronzo Brunetti

**Affiliations:** 1Medical Oncology Unit, “S. Cuore di Gesù” Hospital, 73014 Gallipoli, Italy; antonellalicchetta@libero.it; 2Department of Surgery and Liver Transplantation, Policlinico di Bari, 70124 Bari, Italy; drmemeo@yahoo.it; 3Medical Oncology Unit, National Cancer Research Centre, IRCCS Istituto Tumori “Giovanni Paolo II”, 70124 Bari, Italy; argentieroantonella@gmail.com; 4Department of Biomedical Sciences and Human Oncology, Section of Internal Medicine “G. Baccelli”, University of Bari Medical School, 70124 Bari, Italy; antoniogiovannisolimando@gmail.com; 5Medical Thoracic Oncology Unit, IRCCS Istituto Tumori “Giovanni Paolo II”, Viale Orazio Flacco, 65, 70124 Bari, Italy; vito.longo79@tiscali.it; 6Scientific direction, National Cancer Research Centre, IRCCS Istituto Tumori “Giovanni Paolo II”, 70124 Bari, Italy; delcuratolo.sa@gmail.com

**Keywords:** BRAF, hepatocellular carcinoma, liver microenvironment, MAPK pathway, MEK, Sorafenib, TKI

## Abstract

The few therapeutic strategies for advance hepatocellular carcinoma (HCC) on poor knowledge of its biology. For several years, sorafenib, a tyrosine kinase inhibitors (TKI) inhibitor, has been the approved treatment option, to date, for advanced HCC patients. Its activity is the inhibition of the retrovirus-associated DNA sequences *protein* (RAS)/Rapidly Accelerated Fibrosarcoma *protein* (RAF)/mitogen-activated and extracellular-signal regulated kinase (MEK)/extracellular-signal regulated kinases (ERK) signaling pathway. However, the efficacy of sorafenib is limited by the development of drug resistance, and the major neuronal isoform of RAF, BRAF and MEK pathways play a critical and central role in HCC escape from TKIs activity. Advanced HCC patients with a BRAF mutation display a multifocal and/or more aggressive behavior with resistance to TKI. Moreover, also long non-coding RNA (lnc-RNA) have been studied in epigenetic studies for BRAF aggressiveness in HCC. So far, lnc-RNA of BRAF could be another mechanism of cancer proliferation and TKI escape in HCC and the inhibition could become a possible strategy treatment for HCC. Moreover, recent preclinical studies and clinical trials evidence that combined treatments, involving alternative pathways, have an important role of therapy for HCC and they could bypass resistance to the following TKIs: MEK, ERKs/ribosomal protein S6 kinase 2 (RSK2), and phosphatidylinositol 3-kinase (PI3K)/mammalian target of rapamycin (mTOR). These initial data must be confirmed in clinical studies, which are currently ongoing. Translational research discoveries could create new strategies of targeted therapy combinations, including BRAF pathway, and they could eventually bring light in new treatment of HCC.

## 1. Introduction

Hepatocellular carcinoma (HCC) is a tumor pathology with a strict and predominant resistance to systemic chemotherapy. For this reason, for many years, the disease did not have specific treatment options, which led to poor prognosis [1]. There has been increasing attention given to targeting the key-role genes and pathways. Since 2006, a small molecule inhibitor, sorafenib (Nexavar^®^, Bayer AG, Kaiser-Wilhelm-Allee, Leverkusen, Germania), has been the only approved treatment option, to date, for patients with advanced HCC, giving an overall survival (OS) of 31% (hazard ratio, HR of the sorafenib treated patients, 0.69; *p* < 0.001) [2,3]. Sorafenib inhibits fibroblast growth factor receptor (FGFR) 1, vascular endothelial growth factor receptor (VEGFR) 1–3, c-KIT, and platelet derived growth factor receptor (PDGFR). Moreover, B and Crapidly accelerated fibrosarcoma (RAF) kinases could be inhibited. This interaction lead to inhibition of proliferation, angiogenesis, and activation of apoptosis [4]. After treatment with sorafenib, many alterations in the composition of cytokines, chemokines, and growth factors occur in HCC tissue and blood, with consequent changes in clinical responses [5]. However, its efficacy is hampered by acquired TKI resistance. A great number of data showed that the limited clinical success of these drugs is probably due to the complex relationship between cancer cells and tumor microenvironment in HCC [6,7,8,9]. In this context, another major signaling pathway is being emerged: the mitogen-activated protein kinase (MAPK), responsible of proliferation, migration, and metastasization. Its activity was demonstrated both in the liver “niche” and in the liver microenvironment [10].

## 2. RAS/RAF/MEK/ERK Pathway Role in HCC and Rationale for Targeted Therapies

The most studied and intrigue pathway in HCC is retrovirus-associated DNA sequences*protein* (RAS)/RAF/extracellular-signal regulated kinase (MEK)/extracellular-signal regulated kinases (ERK) pathway. It involve four protein kinases: RAS, RAF, MEK, and ERK. RAS, RAF, and MEK. Also MAPK pathway is activated HCC, such as in several tumors by extracellular signalssich as hormones, growth factors, differentiation factors, and tumor-promoting substances that bond with appropriate receptor tyrosine kinases (RTK) [11,12,13]. After activation, the pathway promotes transcription of genes involved in tumor proliferation. Many data reveal that the somatic gene of phosphoinositide-3-kinase-catalytic-alpha (PIK3CA) result mutated in several human cancer such as HCC [11]. PIK3CA enhances cancer cell proliferation, migration, cancer invasion, and interacts with growth factor-stimulated MAPK signaling [14].

Many studies demonstrated that B-RAF (BRAF) and MEK pathways play a critical and central role in HCC [15,16,17,18]. Initially, Japanese and Chinese studies evidenced that there seems to be scant participation of the BRAF mutations in the etiopathogenesis of HCC [15,16]. However, several recent preclinical studies have demonstrated that the RAS/RAF/MEK/ERK pathway resulted hyperactivated in HCC [17]. If we suggested a molecular treatment approach in HCC, then BRAF pathway would play a crucial and central role in HCC evolution. C-met, a MAPK pathway downstream is often constitutively activated (mediated by BRAF mutation) and this signal regulates cancer cell processes, such as differentiation, proliferation, angiogenesis, and anti-apoptosis [16]. Specifically, MEK and MAPK mRNAs were overexpressed in 40% and 50% of HCC patients, respectively [16]. Also RAF-1 overexpression was present in 100% of HCC patients, significantly high as compared with those with pre-tumoral lesion such as hepatocirrhosis [19]. Furthermore, hepatitis B virus (HBV) and hepatitis C virus (HCV) infections play a crucial role in the activation of the RAS/RAF/MEK/ERK pathway in HCC. Specifically, HCV core protein enhanced the activation of RAF-1 kinase and MAPK/ERK proteins. Moreover, HCC carcinogenesis could be activated through RAS/RAF/MEK/ERK pathway by HCV [20]. Anyway, in a “The Cancer Genome Atlas Program“ (TCGA) study, including 363 HCCs, the prevalence of BRAF mutations was only 0.3% [21]. In another manuscript, using hybrid capture Next-Generation Sequencing (NGS), in 127 HCC patients there were only two BRAF alterations (i.e. one amplification and one non-V600 mutation) [22]. So far, BRAF alteration could to be a potential therapeutic target rather than one of key point in HCC carcinogenesis. Recently, studies have demonstrated a variable prevalence of BRAF mutations in HCC, probably for the difference in geographical origins or racial distributions. Colombino et al., showed that a mutational activation of genes of BRAF and PIK3CA contribute to a more evident HCC tumorigenesis at the somatic level, in the Southern Italian population when compared to other Italian region. Moreover, the same Authors demonstrated that HCC patients with a BRAF mutation are more likely to be multifocal, aggressive, and resistance to TKI therapies [23].

In addition, several studies evidenced a possible influence of the BRAF pathway in the responses of anticancer drugs [24,25]. In HCC, for many years, sorafenib has remained the only small molecular target kinase approved with evident activity against MAPK pathway. Recently, many researches have evidenced that different doses of sorafenib and other TKIs in HCC determine variable results. High drug concentrations produce an inhibition of the MAPK pathway, with the block of proliferation and viability, and the induction of cancer cells apoptosis. Surprisingly, a low dose of TKIs determines an increase of MAPK signaling. These mechanisms could explain the unsuccessful clinical response of the disease to TKIs. We can speculate that the different response rates showed in clinical studies could be due not only to the diverse status of the basal cirrhotic disease, but also to a several putative activated TK due to gene mutations, including BRAF [25]. Several studies have demonstrated that progressive and acquired mutations of HCC during therapies are related with a clinical activity of sorafenib. The phosphorylation decrease of downstream ERK, a putative marker of RAF activity, could be a predictive biomarker of reduced response to sorafenib [26,27].

In recent years, translational research in HCC and other cancers has beenfocused on the study of long non-coding RNA (lnc-RNA), RNAs with over 200 nucleotides in length which provide a regulatory role in different processes such as growth, senescence, differentiation, and apoptosis [28]. The deregulation of lnc-RNAs is implicated in several diseases, including HCC. One of the Inc-RNA involved concerns the BRAF pathway (BRAF Inc-RNA, i.e., BANCR). Much data are available regarding the high levels of BANCR involved in the progression of gastric tumor, colorectal cancer, papillary thyroid carcinoma, and non-small cell lung cancer. Ling Li et al., demonstrated a critical role of BANCR also in the HCC cell line Huh7. They evidenced that BANCR is over-expressed in HCC cells and its downregulation suppresses viability, enhancing apoptosis, and inhibits the migration and invasion of HCC cells. Moreover, downregulation of BANCR inactivates MEK/ERK/N-terminal kinases (JNK), suggesting that these signaling pathways were implicated in the BANC-associated malignance of HCC. BANCR activity could be a possible mechanism of escape by HCC cells to actual therapeutic drugs. Moreover, considering the HCC heterogeneity, the activity of BANCR on other HCC cell lines remains still undefine. However, these data confirmed that BRAF and BANCR system remain potential therapeutic targets in HCC [24].

## 3. Targeting BRAF and MEK in HCC, Preclinical and First Clinical Data

All the data previously described suggest that, even if the RAS/RAF/MEK/ERK pathway plays a central role as potential target in HCC patients, its inhibition could not be sufficient to arrest HCC cells development. A possible strategy could be the combination of RAS/RAF/MEK/ERK pathway inhibitors with other pathways inhibitors. Further clinical studies are mandatory in this setting (Figure 1).

NVP-AAL881 is an oral RAF-1 and VEGFR2 small molecule inhibitor that has been tested in a subcutaneous xenograft model of HCC, with evidence of HCC tumor growth arrest. Specifically, NVP-AAL881 exerts its inhibition inactivating the MEK/ERK pathway and the Signal transducer and activator of transcription 3(STAT3) phosphorylation in a dose-dependent manner [29]. These preclinical data has been confirmed by a recent study of Lang et al., Authors demonstrated the inhibition of HCC xenograft tumors growth after the administration of NVP-AAL881 [30]. Clinical data are necessary to confirm a potential role of this oral RAF inhibitor in advanced HCC patients.

In addition, targeting the MEK pathway became an alternative strategy in HCC patients. Several MEK inhibitors have been submitted in the last years. The first developed MEK1/2 inhibitor was PD98059. Its activity consists in the block MEK1/2 autophosphorylation by “freezing” the MEK pathway into the inactive form. All the subsequently pathway cascade is blocked with arrest of nuclear signal transduction [31]. More recently, a more powerful MEK1/2 inhibitor was discovered, named U0126, and is actually widely used in several experiments [32]. The great limit of these MEK inhibitors is their low solubility and bioavailability. For this reason, the transition from preclinical to clinical studies is still difficult. Actually, we have only one multicenter, open-label, phase 1 trialof a MEK inhibitor (Pimasertib). In this study the drug has been tested in patients with different solid tumors, including HCC. The drug was administered orally at the dose of 45 mg bis in die in 21-day cycles until disease progression or intolerable toxicity [33]. Even if satisfactory data are available regarding the toxicity profile, at the moment a phase II stage of the study has not started [34]. Many other trials are ongoing in this setting [35,36,37].

Furthermore, recent studies evidence that combined treatments have an important role in the treatment of HCC and could be a strategy to overcome the TKIs resistance. Wenhong Wang et al., demonstrated that magnolin, an active ingredient of volatile oil from *Magnolia fargesii* (an oriental medicine to treat nasal congestion and rhinitis), together with BRAF inhibitor SB590885, synergistically suppressed the proliferation of HCC cells. In addition, magnolin suppresses cell migration and invasion through targeting the parallel ERKs/RSK2 signaling pathway, promoting the process of apoptosis [38]. We attend further studies in the clinical setting that could confirm these promising data.

Breunig et al., in a preclinical study, compared sorafenib with the MEK-inhibitors AZD6244 and PD0325901, and the BRAFV600E mutation-specific inhibitor PLX4720 in HCC cell lines regarding the inhibition of MAPK and PI3K to point out a difference of activity in cell proliferation, viability, apoptosis, and chemokine/growth factor secretion [39]. They demonstrated that MAPK inhibitors significantly alter the HCC microenvironment and confirmed that BRAF and MEK inhibitors display dose-dependent antiproliferative and proapoptotic activities. Specifically, the drug response is highly dependent on the concentration of the drugs, as well as on the individual susceptibility of the HCC cells. The secretion of chemokines is also influenced by different doses of MEK and BRAF inhibitors. They inhibit the secretion of interleukin-8 (IL-8) and VEGF and exert an influence on tumor microenvironment with the arrest of proangiogenic activities (tumor escape concept) and the development of metastases [39]. A clinical study is currently being conducted with a RAF inhibitor in patients with several metastatic solid cancer, including HCC. It’s activity appears to be able to modulate both the MEK/ERK and PI3 /Akt/mTOR pathways [35].

Kim et al., showed that a dual PI3K/mTOR inhibitor, named CMG002, in combination with sorafenib significantly inhibits HCC cells proliferation and metastasis in in vitro and in vivo experiments. Also this activity is exerted by inhibiting both RAS/RAF/MAPK and PI3K/AKT/mTOR pathways. This study confirmed the optimistic strategy of combined TKIs inhibitors in HCC treatment [40].

## 4. Conclusions

The success of the treatment of advanced HCC patients goes through a best understanding of the HCC pathogenesis. In this context, the MAPK pathway and BRAF signaling play a crucial role in the regulation of HCC cell proliferation and survival. Notwithstanding the great relevance of this complex-signaling network, we have not yet fully understood how its inhibition could generate a very benefit in HCC treatment. TKIs inhibitors exert their activity against HCC cells inhibiting BRAF signaling, however, many resistances occurred under treatment with tumor escape. Actually, Sorafenib remains the unique TKI for the treatment of HCC able to block signal transduction mediated by the MAPK pathway, however, HCC cells often presented an intrinsic and acquired resistance to TKIs, due to several mutations and loss of functions [41]. The underlying resistance mechanisms are actually unknown. We need of specific diagnostic tests to predict HCC response to BRAF-MEK inhibitors. Nonetheless, novel insights suggested significant investigation in solid and hematological malignancies, with initial with promising results [42,43]. Recently, several data confirmed that different doses of TKIs determine inhibition or iper-activaction of the MAPK pathway, with opposite effects. In addition, translational research in HCC showed a critical role of BANCR in the acquired resistance of MAPK pathway to TKIs. Finally, the HCC cells overcome the pharmacological inhibition by activating alternative pathways. The key challenge for BRAF pathway inhibition will likely be the level of cross talk and negative feedback along parallel pathways. A possible strategy could be the combination of RAS/RAF/MEK/ERK pathway inhibitors with other pathways inhibitors. Further clinical studies are mandatory in this setting. For this reason, it will be fundamental to identify any predictive molecular response factors in order to customize the treatment of a chameleon-like disease such as HCC [44]. Recent breakthroughs could propose the strategy of combined targeted therapy in HCC, including BRAF pathway. The multi-pathways inhibition, together with more frequent customized treatments, could eventually open the door for better results in the treatment of HCC.

## Figures and Tables

**Figure 1 medicina-55-00754-f001:**
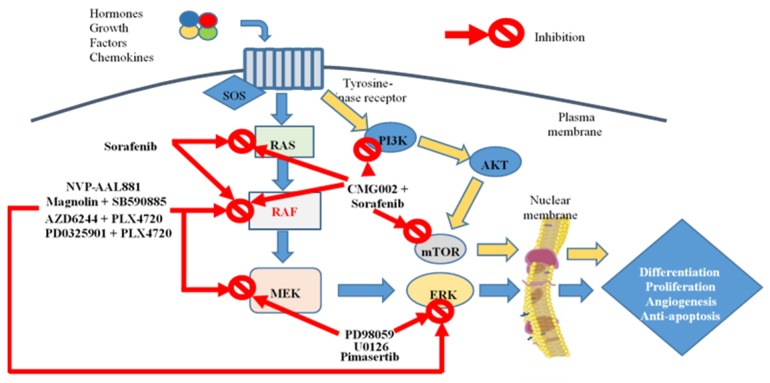
RAS/RAF/MEK/ERK and PI3K/AKT/mTOR pathways: current tyrosine kinase inhibitors (TKIs) under clinical trials. Abbreviations—ERK: extracellular-signal regulated kinases; MEK: mitogen-activated and extracellular-signal regulated kinase; mTOR: mammalian target of rapamycin; RAF: Rapidly Accelerated Fibrosarcoma *protein*; RAS: retrovirus-associated DNA sequences *protein*; PI3K: phosphatidylinositol 3-kinase.

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
