# Peer review of "Role of BRAF in Hepatocellular Carcinoma: A Rationale for Future Targeted Cancer Therapies"

_1010-660X, 2019, doi:10.3390/medicina55120754_

Round 1

Reviewer 1 Report

Article provides a good overview of the role of the Ras/Raf/MEK/ERK pathway as well as the preliminary data and ongoing clinical trials.  The information appears to be comprehensive and I do not have any recommendations for modification for the authors

Author Response

We thank the first reviewer for his appreciation of the work. We provide minor language revision as required.

Reviewer 2 Report

I congratulate the authors for this review of the role of BRAF in HCC. Its major strength is the concise description of MAPK pathway in HCC including potential targets for therapy. However, I think there are relevant issues to address:

Major comments:

While this review is well written and the authors are well versed in the significance of MAPK pathway in HCC, the review fails to demonstrate the relevance of BRAF in this context.

The authors do not address that BRAF alterations are uncommon in HCC since the results obtained in reference 21 have not been replicated. For example, in the TCGA paper that included 363 HCCs, the prevalence of BRAF alterations was 0.3% (Cell. 2017 Jun 15;169(7):1327-1341) and other report using hybrid capture NGS in 127 samples there was only two BRAF alterations (BRAF amplification and BRAF non-V600 mutation) (Clin Cancer Res. 2019 Apr 1;25(7):2116-2126).

If authors can not demonstrate the specific relevance of BRAF in the context of MAPK pathway, I recommend to change the focus to potential targets in the pathway rather in BRAF gene.

Minor comments:

- In order to provide context it would be better to mention sorafenib´s overall survival benefit instead of the absolute survival.

- The statement: “In HCC, for many years, sorafenib has remained the only small molecular target kinase approved and the most successful anti-BRAF inhibitor.” would not be accurate. In the study REFLECT lenvatinib showed non-inferiorty to sorafenib with better PFS and RR. Furthermore, both sorafenib and regorafenib have anti BRAF activity, however, it is not clear that this is a relevant mechanism of action in HCC. There are more specific BRAF monomer inhibitors such as dabrafenib, vemurafenib and encorafenib which do not have currently a role in HCC.

Author Response

Major comments:

Reviewer 2 suggests a modification in the text about the real potential target in HCC: BRAF genes instead of BRAF pathway (“The authors do not address that BRAF alterations are uncommon in HCC since the results obtained in reference 21 have not been replicated. For example, in the TCGA paper that included 363 HCCs, the prevalence of BRAF alterations was 0.3% (Cell. 2017 Jun 15;169(7):1327-1341) and other report using hybrid capture NGS in 127 samples there was only two BRAF alterations (BRAF amplification and BRAF non-V600 mutation) (Clin Cancer Res. 2019 Apr 1;25(7):2116-2126). If authors can not demonstrate the specific relevance of BRAF in the context of MAPK pathway, I recommend to change the focus to potential targets in the pathway rather in BRAF gene”). We thank the reviewer and we agree with the requested clarification. Therefore, the paragraph is modified (highlighted in yellow in the text) and the two bibliographical notes correctly reported by the reviewer are added.

Minor comments:

In order to provide context it would be better to mention sorafenib´s overall survival benefit instead of the absolute survival. We have changed this data in the manuscript (see in the text)

- The statement: “In HCC, for many years, sorafenib has remained the only small molecular target kinase approved and the most successful anti-BRAF inhibitor.” would not be accurate. In the study REFLECT lenvatinib showed non-inferiorty to sorafenib with better PFS and RR. Furthermore, both sorafenib and regorafenib have anti BRAF activity, however, it is not clear that this is a relevant mechanism of action in HCC. There are more specific BRAF monomer inhibitors such as dabrafenib, vemurafenib and encorafenib which do not have currently a role in HCC. We have corrected the statement as requested (we take off the expression “most successful anti-BRAF inhibitor, effectively inadequate). See in the text the evidenced modified form. Many thanks to the revisor for the observation.

Reviewer 3 Report

Please find my comments in the attached file.

Author Response

We provide language revision as required.

“Furthermore, for first time, translational research analyzed the long non-coding RNA concerns the BRAF pathway as possible mechanism of escape by HCC cells to actual TKIs. This sentence from the abstract is unclear, should be verified”. We have corrected the sentence in abstract as requested (see the evidenced form in abstract).

In the introduction section I cannot agree with the statement: Hepatocellular carcinoma (HCC) is a tumor disease with a strict and predominant resistance to systemic chemotherapy. For this reason, for many years the disease did not have specific treatment options, which led to poor prognosis. The resistance to chemotherapy is not the reason for the absence of specific treatment but rather the lack of specific targets. We have corrected our inaccurate expression as requested (see the evidenced form in the introduction section).

We can speculate that the different response rates observed in the clinical trials could be due not only to the different status of an accompanying cirrhotic disease, but also to a different distribution of candidate gene mutations, including BRAF. In this sentence it is not clear what ‘different distribution of gene mutations” means, it should be specified. As requested, we explain the sense of the sentence with the addition of results of two pubblications (see the manuscript in the section and in bibliography, references 26 and 27)

Similarly, the sentence from the page 3 is not clear: These data could explain that the only suppression of MEK/ERK and MAPK signaling pathways using chemotherapeutic drugs has not yielded great and expected improvements in the management of HCC. We agree with revisor and we have removed the sentence from the manuscript.

Round 2

Reviewer 2 Report

I have no further comments.